# Controllable Fabrication of Gallium Ion Beam on Quartz Nanogrooves

**DOI:** 10.3390/mi15091105

**Published:** 2024-08-30

**Authors:** Peizhen Mo, Jinyan Cheng, Qiuchen Xu, Hongru Liu, Chengyong Wang, Suyang Li, Zhishan Yuan

**Affiliations:** 1School of Electro-Mechanical Engineering, Guangdong University of Technology, Guangzhou 510006, China; 3219000411@mail2.gdut.edu.cn (P.M.); 3221000057@mail2.gdut.edu.cn (J.C.); 3121000052@mail2.gdut.edu.cn (Q.X.); 3223000265@mail2.gdut.edu.cn (H.L.); cywang@gdut.edu.cn (C.W.); 2Guangdong Provincial Key Laboratory of Minimally Invasive Surgical Instruments and Manufacturing Technology, Guangdong University of Technology, Guangzhou 510006, China; 3State Key Laboratory for High Performance Tools, Guangdong University of Technology, Guangzhou 510006, China; 4Smart Medical Innovation Technology Center, Guangdong University of Technology, Guangzhou 510006, China

**Keywords:** focused ion beam (FIB), layer-by-layer etching, V-shaped nanogrooves, etching process parameters

## Abstract

Nanogrooves with high aspect ratios possess small size effects and high-precision optical control capabilities, as well as high specific surface area and catalytic performance, demonstrating significant application value in the fields of optics, semiconductor processes, and biosensing. However, existing manufacturing methods face issues such as complexity, high costs, low efficiency, and low precision, especially in the difficulty of fabricating nanogrooves with high resolution on the nanoscale. This study proposes a method based on focused ion beam technology and a layer-by-layer etching process, successfully preparing V-shaped and rectangular nanogrooves on a silicon dioxide substrate. Combining with cellular automaton algorithm, the ion sputtering flux and redeposition model was simulated. By converting three-dimensional grooves to discrete rectangular slices through a continuous etching process and utilizing the sputtering and redeposition effects of gallium ion beams, high-aspect-ratio V-shaped grooves with up to 9.6:1 and rectangular grooves with nearly vertical sidewalls were achieved. In addition, the morphology and composition of the V-shaped groove sidewall were analyzed in detail using transmission electron microscopy (TEM) and tomography techniques. The influence of the etching process parameters (ion current, dwell time, scan times, and pixel overlap ratio) on groove size was analyzed, and the optimized process parameters were obtained.

## 1. Introduction

Nanogrooves possess unique physical and chemical properties, demonstrating significant application value in modern technology. In terms of their physical properties, they exhibit a size effect that is smaller than the wavelength of visible light, resulting in macroscopic changes in optical, electrical, and other properties. This capability enables high-precision control of light with exceptional optical and magnetic properties. In the field of optics, V-shaped grooves can be utilized for integrating hybrid dielectric waveguides and for accurately aligning nanowires, thereby enhancing mode-coupling efficiency and field enhancement [1]. In semiconductor integrated circuit processes, V-shaped grooves are utilized in VMOS transistors, thereby enhancing integration density [2]. In the field of semiconductor photovoltaics, silicon wafers with V-groove structures were used in solar cells to improve light conversion efficiency [3]. In terms of chemical properties, nanogrooves possess a high specific surface area and catalytic performance, exhibiting excellent chemical activity and stability, thereby inducing extremely high chemical reactivity. In the field of biosensing, V-shaped electrode arrays demonstrate significant potential in the collection and transmission of biological signals [4]. Rectangular metal gratings can achieve dual-channel absorption enhancement of graphene in the visible light spectrum, thereby enhancing its detection capabilities [5]. Nano-rectangular gratings can be utilized for highly sensitive detection of hydrogen concentration and spectral sensing [6]. Nano-rectangular grooves, when applied in magnetic tunnel junctions (MTJs) and giant magnetoresistive (GMR) devices, are expected to enhance magnetoresistance through the coupling of Coulomb and spin blockade [7]. Nano-rectangular grooves integrated into surface plasmon resonance (SPR) sensors are expected to exhibit significantly higher differences in reflected signals, due to local changes in refractive index, compared to conventional SPR sensing systems [8]. The manufacturing methods for nanogrooves have always been a focal point of interest in the scientific research field.

Currently, the preparation methods for nanogrooves are primarily divided into three major categories: mechanical processing, chemical etching, and high-energy beam processing. In terms of mechanical processing, the methods mainly include cutting, fly cutting, abrasion, and electrical discharge machining. Zhang and colleagues used a cutting method that leverages microcracks to fabricate V-shaped grooves [9]. The resulting V-grooves had average angles of 61.7°, 91.8°, and 119.4°, with corresponding heights of 545 µm, 495 µm, and 479 µm, respectively. This method achieved significant results at the microscale, but when scaled down to the nanoscale, direct application becomes difficult. Li used a PCD (polycrystalline diamond) turning tool to perform fly cutting on aluminum material to create V-shaped grooves [10], achieving a V-groove angle of 72.2°, a groove bottom width of 35.03 µm, and a groove depth of 51 µm. However, as the processing dimensions enter the nanometer scale, tool wear and heat accumulation issues become more pronounced, limiting the application of fly cutting technology in the preparation of nanogrooves. Guo and colleagues used electrical discharge machining to create an array of V-shaped grooves on 316 stainless steels, with widths of 1 mm and depths of 0.4 mm [11]. However, at the nanoscale, the energy density and stability of the electrical discharge are difficult to control, leading to difficulties in ensuring processing accuracy and surface quality.

The methods for fabricating nanogrooves by chemical etching mainly include two types: one is wet etching, the chemical removal of material on the surface of a substance using liquid chemical reagents. The other is dry etching, referring to the use of a sufficiently large electric field applied to a gas to break it down and generate plasma. The plasma passes through the window opened in the photoresist and physically or chemically reacts with the substance to remove the surface material. Shi and colleagues employed a combination of lithography and wet etching to fabricate nanogrooves [12]. Initially, they used ion beam etching to create a silicon nitride mask, followed by etching with KOH on a silicon (100) surface for 8 min to fabricate V-shaped channels with a width of 200 nm and a bottom angle of 54.78°. This method offers the advantages of good repeatability and the potential for large-scale production, but it has limited processing range and can only produce inverted conical grooves with fixed bottom angles under specific crystallographic orientations.

Ren and colleagues utilized inductively coupled plasma (ICP) etching to process GaN material [13]. They etched continuous V-shaped steps, 1.837 μm in depth, using photoresist as a mask and controlled the etching angle within 75°. This method exhibits strong anisotropy and high selectivity, but it can cause significant damage to the material surface, present contamination issues, and can struggle with forming finer patterns [14].

High-energy beam processing mainly includes laser etching, electron beam etching, and ion beam etching. Laser etching is a technique that uses the interaction of a laser beam with materials to remove material through mechanisms such as photothermal effects, photochemical effects, or photomechanical effects. Electron beam etching uses a focused electron beam to bombard the material surface, achieving material removal through collisions and energy transfer between electrons and material atoms. Focused ion beam (FIB) etching is a technique that uses a focused ion beam to bombard the material surface, achieving material removal through collisions and energy transfer between ions and material atoms.

Zhang and colleagues used femtosecond laser single-step etching to create a V-shaped fiber microcavity [15], with the distance between the two end faces of the V-microcavity core being 6 μm, the microcavity depth 86 μm, and the base angle 7.17°. This method has the advantages of a fast processing speed, small heat-affected zone, and minimal contamination, but the processing accuracy and depth of laser etching can be influenced by various factors such as the quality of the laser beam, material properties, and processing environment. Dwir and colleagues used electron beam etching to manufacture V-shaped grooves that were 100 nm wide and 1.5 μm deep [16]. This method offers high resolution and flexibility but has a relatively lower yield.

Kim et al. conducted focused ion beam (FIB) experiments using an FEI Strata DB-235 dual-beam FIB system [17]. The samples consisted of an epitaxially grown 100 nm thick GaAs layer on an n-GaAs substrate, with a 50-angstrom-thick titanium layer deposited on the surface to prevent charging effects during electron or ion beam exposure. They successfully fabricated nanostructures with a diameter of 80 nm and a hole depth of 125 nm. Li et al. utilized a dual-beam scanning electron microscopy (SEM)/focused ion beam (FIB) system to pattern fused silica substrates coated with a thin Cr layer [18]. They successfully created nano-gratings with a width down to 54 nm and an aspect ratio greater than three, demonstrating the capabilities of FIB technology in micro-/nano-patterning. Choi employed specific processes and FIB etching methods to obtain nano-V-grooves, with the side angles of the grooves measuring 16.8°, the lower end spacing being approximately 81 nm, and the minimum width reaching 55 nm [19]. Kannegulla and Cheng utilized focused ion beam etching (FIB) to create V-shaped grooves [20] on a silicon substrate where aluminum (Al) and silver (Ag) had been deposited. The FIB milling was conducted using an FEI Quanta 3D FEG, adjusting the dosage and current of the gallium ion beam to achieve V-shaped channels measuring 100 nm wide and 200 nanometers deep. This technique provides high processing precision, is free of contamination, and allows for maskless processing. However, controlling the size and morphology of the nanogrooves is challenging, and non-conductive materials are prone to drift phenomena. Nonetheless, it remains the sole method for processing nanogrooves of extremely small dimensions.

The layer-by-layer etching process provides an advanced nanofabrication method characterized by the alternating deposition and removal of atomic layers on the surface of materials. This technique exhibits notable advantages over the conventional focused ion beam (FIB) etching technology. Initially, the etching rate of layer-by-layer etching significantly exceeds that of traditional methods [21], attributable to the capability of each etching cycle to remove multiple monolayers simultaneously. Subsequently, the precision of etching is markedly enhanced, enabling more accurate control of layer thickness and quantity. Additionally, the layer-by-layer etching minimizes material damage caused by ion beams, as it employs a reduced energy level during each etching step. This approach also affords effective control over the etching depth and profile, thus avoiding the issues of rough sidewalls and uneven bases that are prevalent in standard FIB etching procedures.

To overcome the issue of difficulty in precisely controlling the size and morphology when etching nanogrooves with single-pixel point etching in traditional focused ion beam (FIB), this paper proposes a novel FIB etching technology based on a layered etching process. The study specifically includes research on the layered etching process, simulation of the ion beam sputter etching process, TEM analysis of the morphology and composition of the sidewalls of the V-shaped grooves, and research on the impact of the process parameters on the size of the nanogrooves. The study successfully fabricated V-shaped and rectangular nanogrooves on silicon dioxide material using a focused gallium ion beam. It further simulated the ion beam sputter etching process and conducted an in-depth analysis of the morphology and composition of the V-groove sidewalls using transmission electron microscopy (TEM) tomography. Additionally, it discussed the specific impact of factors such as ion current, dwell time, number of scans, and pixel overlap rate on the size of the nanogrooves. This is beneficial for addressing the issues where the formation mechanism and aspect ratio control methods of nanogrooves in other FIB etching techniques are not yet clear, offering new perspectives and approaches for the advancement of the nanomanufacturing field.

## 2. Simulation of FIB Sputter Etching Process

In ion beam etching, the surface structure of the material affects the sputtering yield and etching rate, and the redeposition effect complicates the surface structure [22]. This complexity leads to significant variations in the shape of etching under the same incident dose due to minor process differences [23]. This article focuses on the simulation of sputtering processes during ion beam etching, studying the impact of the redeposition effect on the structure of nanogrooves prepared by the layer-by-layer etching process.

### 2.1. FIB Sputter Etching Process Simulation Methods

This study refers to Refs. [24,25] and conducts an in-depth analysis of the focused ion beam (FIB) etching process based on the ion sputtering flux and redeposition model, incorporating the cellular automaton algorithm.

In the simulation process, the substrate of the FIB system is first divided into a grid composed of square cells, each with a set of parameters. The cells are categorized into three states: “surface cells” are those that may be sputtered or grown, which must be material cells with at least one neighbor being an air cell; “material cells” are those that have not participated in sputtering or deposit, and whose inherent parameters remain unchanged; “air cells”, on the other hand, are cells that have had their inherent parameters reduced to zero after sputtering and deposition processes. When determining the surface cells, the 4-neighbor method is used, which considers the four directly adjacent neighbors in front, behind, to the left, and to the right of the cell. The geometric shape of the pattern is defined by the size of the cells, their initial state, and the number of them in both the horizontal and vertical directions.

The dose of the focused ion beam is approximately distributed according to a Gaussian intensity profile, with the center of the ion beam having a higher intensity than the surrounding areas. This distribution results in a stronger etching effect and deeper etching depth at the center of the ion beam, forming the etching contour shown by the dashed line in Figure 1. The Gaussian distribution function of the ion beam energy is
(1)$$J\left(x,y\right)=\frac{I}{2\pi \sigma^{2}}\exp\left(-\frac{x^{2}+y^{2}}{2\sigma^{2}}\right)$$

σ represents the standard deviation, which is related to the half-maximum width $d_{m}$ of the ion beam distribution,$\;\sigma =\frac{d_{m}}{\sqrt{8ln2}}$; I represents the ion beam current; with x and y being the scanning strategy functions that are dependent on time t. The formula discussed here is derived from the referenced literature and has been appropriately simplified [25].

The incident ion flux of the ion beam is
(2)$$F_{inc}=\frac{I}{2\pi e\sigma^{2}}exp(-\frac{x^{2}+y^{2}}{2\sigma^{2}})$$

A large number of high-energy particles interact with the substrate material surface, causing sputtering of surface particles, and the sputtered atom flux is
(3)$$F_{dir}=Y_{\left(\theta \right)}F_{inc}cos\theta$$

$Y_{\left(\theta \right)}$ represents the sputtering yield at the ion beam incidence angle θ. The formula discussed here is derived from Ref. [26].

The sputtered particles may interact with the substrate surface again, undergoing sputtering or redeposition; the redeposition flux is
(4)$$F_{redep}=F_{dir}\frac{dA}{A}$$

A represents the total range of sputtered atoms; dA represents the range of deposited atoms.

Based on the geometric relationship in Figure 1, the following can be calculated:$$\frac{sin\gamma }{2}=\frac{dl}{2d},\;A=2\int_{0}^{\frac{\pi }{2}}cosx\;dx,\;dA=\int_{\alpha -\frac{\gamma }{2}}^{\alpha +\frac{\gamma }{2}}cosx\;dx$$

d is the distance between two cells, dl is the edge length of a cell, γ is the emission angle corresponding to the cell, and α is the emission angle of the sputtered atoms.

Then,
(5)$$\;F_{redep}=F_{dir}f_{\alpha }\frac{dl}{2d}$$

$f_{\alpha }$ is the angular distribution function of the sputtered atoms.

The results from Monte Carlo calculations indicate that $f_{\alpha }$ can be fitted to the power of a cosine function, with n typically ranging from 1 to 3. Sun and colleagues found through simulation that when n is set to 1, it can effectively describe the angular distribution of sputtered ions during FIB system processing [25].

The simulation procedure for this project is as follows: First, scan the cells within the simulation area, identify the surface cells, and then, calculate the sputtered atom flux, taking into account the redeposition effect and updating the status. Next, summarize the total particle flux for each cell, identify the last cell, and determine whether it is a deposit or an air cell based on whether the total particle flux of the cell is greater than zero. Finally, verify whether the etching has reached the specified depth; if not, continue to scan the next layer and repeat the process, ultimately outputting a two-dimensional graphic of the etching results. The flowchart of the simulation program is shown in Figure 2.

### 2.2. Focused Ion Beam Layer-by-Layer Etching of V-Groove Simulation

This project uses Visual Studio 2019 to write the simulation code, with simulation parameters set through the MFC (Microsoft Foundation Classes) interface; SiO_2_ is used as the substrate with an atomic density of 2.2 × 10^10^ atoms per µm^3^. The SiO_2_ is bombarded by a gallium ion source from a focused ion beam (FIB), with the ion beam voltage set to 30 kV, corresponding to an ion energy of 30 keV. During the sputtering process, the sputtering yield is the number of target atoms sputtered by an incident high-energy charged particle, which can be used to measure the sputtering rate. Under these conditions, by referring to Ref. [27], it can be determined that the sputtering yield for gallium ions incident vertically on SiO_2_ is approximately 2. Therefore, the ion beam is incident perpendicularly on the sample surface to provide the maximum incident energy. Based on the sputtering yield of 2 under this condition [28], the sputtering yield is accordingly set to 2. Due to the use of a small beam current, the irradiation damage to the sample surface can be reduced when imaging with the ion beam, and the small-beam-current ion beam has a small half-peak width diameter, which is beneficial for the preparation of fine nanostructures. Therefore, the ion beam current is set to 5 picoamperes (pA), corresponding to an ion beam diameter of 10 nanometers (nm) under this beam current. The reset rates for both the X and Y directions are set to 50% to balance the uniformity of etching. To control the simulation time, the length and width of the V-shaped groove are set to 200 nm, and the etching is based on a cross-section of 200 nm width for layer-by-layer etching. The rectangular slices are set to widths of 200 nm, 180 nm, 160 nm, 140 nm, 120 nm, 100 nm, 80 nm, 60 nm, 40 nm, and 20 nm, totaling 10 layers. The total etching time for each layer is set to 20 s. In the simulation, the substrate’s dimensions should be greater than or equal to the dimensions of the etched groove, so the length, width, and height of the substrate are set to 200 nm, 500 nm, and 3000 nm, respectively. The size of the cells is set to be smaller than the diameter of the ion beam to ensure the accuracy of the simulation. The model in this paper is based on the assumption that a particle interacts with only a single cell. According to Sigmund’s sputtering theory [29], and considering both the simulation accuracy and computational efficiency, the cell size is set to 8 nm by 8 nm by 8 nm.

After setting the above parameters, the etching process begins. Table 1 displays the experimental parameters for the layer-by-layer etching of rectangular trenches to form V-shaped grooves. Each row of data represents the data for each etching pattern. Once the simulation reaches the specified etching depth, the simulation results are read using the MATLAB 9.9 2020 software to plot three-dimensional mesh diagrams to display the etching structure. Figure 3a–f are the three-dimensional surface diagrams of the V-groove etched for two, four, six, seven, nine, and ten layers, respectively. The simulation results show that the widths of the V-grooves etched for two, four, six, seven, and nine layers are 200 nm, with depths of 500 nm, 1000 nm, 1500 nm, 1750 nm, and 2250 nm, respectively, which are roughly in line with expectations. The final result of this simulation is shown in Figure 3f, where the V-groove has a width of 200 nm and a depth of 2504 nm, resulting in a depth-to-width ratio of 12.5:1. The shape is approximately an inverted triangle with sharp base angles.

## 3. Experiments of Focused Ion Beam (FIB) Etching Nanogroove Fabrication

### 3.1. Experimental Material and Equipment

This experiment selects quartz glass with a thickness of 5 mm as the etching material to fabricate V-grooves and rectangular grooves, and to study their machining conditions. However, since quartz is a non-conductive material, direct processing with a focused ion beam (FIB) can lead to charging accumulation on the surface, causing drift phenomena that affect the machining accuracy. Thus, prior to the experiment, a 15 nm to 30 nm thick layer of Au was deposited on the quartz glass as a conductive layer using a sputtering coater (Japan Vacuum Instruments Ltd., Mito, Japan, MSP-1S). Considering the thin nature of the conductive gold layer and the substantial disparity in thickness relative to the quartz glass, any impact it may have on the local electric field distribution and the angles of incidence of the incoming ions is deemed negligible.

The V-groove and rectangular groove fabrication experiments in this study were both conducted using the LYRA 3 XMU focused ion beam–field-emission scanning electron microscope (FIB-SEM system) from Tescan, a company in the Bmo, Czech Republic. This system offers secondary electron resolution as high as 1.0 nm, allowing for the acquisition of images with high resolution, high contrast, and low noise. The metal ion source is Ga+, with an ion current ranging from 1 pA to 50 nA and an acceleration voltage of 500 V to 30 kV. At an acceleration voltage of 30 kV and a working distance of 9 mm, the ion beam can achieve a resolution of 2.5 nm.

### 3.2. Experimental Process

#### 3.2.1. Fabrication of V-Grooves

The etching of V-grooves is broken down into multiple figures with gradually decreasing widths, where the minimum width is *b*, the difference in width between each pair of layers is *c*, and the depth of all figures is *h*. The width of the figure with the maximum width is *b* + 2 × (*i* − 1) × *c*, and the total depth of the V-groove is *I* × *h*. This experiment uses gallium ion beams as an example, setting the type of ion beam for the focused ion beam−field−emission scanning electron microscope, with a voltage range from 0.5 kV to 30 kV and a current range from 1 pA to 10 nA. Since the energy distribution of the ion beam is similar to a Gaussian distribution, with the center intensity higher than the surrounding areas, the etching effect is enhanced with an increase in ion beam intensity, thus forming a V-groove under continuous layer-by-layer etching. The FIB layer-by-layer etching process is shown in Figure 4. During this process, the focused ion beam first etches the Au layer, and then, etches layer by layer according to the designed decomposed figures, with the same depth between adjacent layers and a gradually reduced width. By etching different numbers of layers, changes in the contour of the V-groove can be observed. Figure 4 depicts the schematic contour diagrams of V-grooves etched into silicon dioxide after 1, 4, 7, and 10 layers with an ion beam. The simulation employs a serpentine scanning method, which is designed to closely resemble actual experiments, as this type of scanning is commonly utilized in real-world equipment during testing.

When preparing V-grooves using the LYRA 3 XMU focused ion beam–field-emission scanning electron microscope, the process path follows the same path as in the simulation. First, the glass sample coated with a gold layer (with a thickness of 20 nm for Au and 3 mm for SiO_2_) is placed into the sample chamber, and the vacuum level is maintained at 10^−7^ mbar. After focusing the electron beam, the sample stage is adjusted to a working distance of 9 mm and rotated by 55° to be perpendicular to the ion gun. The acceleration voltage of the ion gun is set to 30 kV and the ion current to 5 pA, at an overlap rate of 50% in both the X and Y directions. In the DrawBeam 4.2.35.0 software, a V-groove is drawn, composed of 10 rectangles with widths decreasing from largest to smallest (the largest width being 200 nm and the length being 2 µm), with each rectangle having a width difference of 20 nm, and the corresponding etching times are set. Subsequently, the ion beam is activated to perform continuous micro-area processing and etching on the glass sample, avoiding the use of ion beam imaging and real-time observation functions (FIB observer) to prevent contamination and etching deviation. Following the etching process, the sample surface morphology is analyzed using the electron beam imaging function. Table 2 lists the experimental parameters for the layer-by-layer etching of rectangular trenches to form V-shaped grooves.

The surface morphology and cross-sectional morphology of the sample are shown in Figure 5, with the SEM scale bars in the figures being 1 μm for both the surface and cross-section. Multiple V-grooves of varying depths were prepared through layer-by-layer etching. Specifically, after etching for 4 layers, the width was 242 nm and the depth was 388 nm; for 6 layers, the width was 247 nm and the depth was 912 nm; for 8 layers, the width was 245 nm and the depth was 1208 nm; and for 10 layers, the width was 175 nm and the depth was 1680 nm. These dimensions are close to the simulated profiles, but there are slight differences in surface width, which is related to the focusing stability of the FIB gallium ion beam during the actual etching process. The high-voltage power supply used in the focused ion beam system determines the performance of the ion optical system, and its stability largely affects the size and stability of the beam spot [30]. The FIB system requires high focusing stability during the etching process to ensure the precise distribution of the ion beam on the sample surface. If the focusing stability is poor, it may lead to variations in the diameter of the ion beam spot, thereby affecting the etching width and depth.

The actual depth of the V-grooves fabricated may vary from the simulated values, primarily due to the effect of redeposition. During FIB etching, high-energy ions interact with the sample, causing material atoms to be sputtered from the surface. Most of these sputtered atoms are evacuated by the vacuum pump, but some atoms are redeposited on the sample surface or the sidewalls of the structure. As the etching depth increases, the evacuation efficiency of the vacuum pump for these sputtered atoms decreases, making the redeposition effect more pronounced. This redeposition phenomenon forms an amorphous layer in the etching area, affecting the etching depth and the verticality of the sidewalls, resulting in the actual etching depth being less than the depth predicted by the simulation. The SEM images show that the conical groove structure is clear, with the nanogroove cross-section approximately taking the shape of an inverted triangle, having a sharp apex and smooth sidewalls without a serrated step structure. Contrary to the simulation that showed layering, the secondary ion sputtering redeposition from the ion beam etching filled in the intervals of the layered etching. This indicates that FIB layer-by-layer etching can achieve the fabrication of V-grooves.

#### 3.2.2. Fabrication of Rectangular Grooves

When etching a rectangular groove, the target groove is divided into i figures with the same width, and the etching time is set to increase incrementally. On the focused ion beam–scanning electron microscope, a gallium ion beam is used for etching by adjusting the voltage (0.5 kV to 30 kV) and the current (1 pA to 10 nA). The FIB layer-by-layer etching process is depicted in Figure 6. Initially, the Au layer is etched, followed by etching layers of the same width according to the decomposed figures, with the etching time increasing incrementally for each layer. By etching different numbers of layers, the changes in the contour of the rectangular groove can be observed. The results of the rectangular groove contour after etching silicon dioxide for 1, 3, and 5 layers with an ion beam are shown in Figure 6. The etching of the rectangular groove is consistent with the etching of the V-groove in terms of the scanning path; the difference lies only in the settings for the etching layers.

When using the LYRA 3 XMU focused ion beam–field-emission scanning electron microscope to fabricate rectangular grooves of different widths on a silicon dioxide substrate, the pre-etching operational steps and parameter settings are the same as those described previously for the fabrication of V-grooves. Rectangular grooves with a length of 2 μm and widths of 200 nm, 400 nm, 600 nm, and 800 nm are etched. In the DrawBeam software, three rectangles of the same width are drawn to represent the slices of each three-dimensional nanogroove, with etching times set at 40 s, 60 s, and 80 s. Following the etching process, the surface morphology and cross-section of the sample are observed using the electron beam and ion beam functions. The entire process is conducted using the dual-beam electron microscopy system.

Figure 7’s SEM images allowed us to measure the depth and width of the rectangular grooves etched in the experiment. For the structure with a designed width of 200 nm, the actual width after etching was 282 nm and the depth was 1072 nm, with the sidewalls inclined at 12°; for the structure with a designed width of 400 nm, the actual width after etching was 446 nm and the depth was 1000 nm, with the sidewalls inclined at 4.2°; for the structure with a designed width of 600 nm, the actual width after etching was 743 nm and the depth was 405 nm, with the sidewalls inclined at 3.1°; for the structure with a designed width of 800 nm, the actual width after etching was 892 nm and the depth was 396 nm, with the sidewalls inclined at 2.8°. Detailed experimental parameters and data can be found in Table 3.

The surface morphology and cross-sectional morphology of the sample are shown in Figure 7, with the SEM scale bars in the images being 1 μm for both the surface and the cross-section. Multiple rectangular grooves of varying depths were prepared through layer-by-layer etching. For the structure with a set etching width of 200 nm, the resulting rectangular groove was 282 nm wide and 1072 nm deep, with sidewalls inclined at an angle of 12°; for the structure with a set etching width of 400 nm, the resulting rectangular groove was 446 nm wide and 1000 nm deep, with sidewalls inclined at an angle of 4.2°; for the structure with a set etching width of 600 nm, the resulting rectangular groove was 743 nm wide and 405 nm deep, with sidewalls inclined at an angle of 3.1°; for the structure with a set etching width of 800 nm, the resulting rectangular groove was 892 nm wide and 396 nm deep, with sidewalls inclined at an angle of 2.8°. The surface width of the rectangular grooves differs from the simulated width for the same reasons as the fabrication of V-grooves. From the cross-sectional SEM images, it can be seen that during etching of the narrow width (200 nm), the sidewall angle is larger due to ion deposition on the sidewalls and bottom over a long etching period. As shown in Figure 7e, when the total etching time is the same, the sidewall angle decreases and approaches perpendicularity as the etching width increases.

### 3.3. Morphology and Composition of V-Groove Sidewalls

This study employs a high-resolution transmission electron microscope (FIE TalorsF200X) to analyze the sidewalls and composition of the V-grooves. Following the fabrication process for V-grooves detailed in the preceding section, new structures were produced in additional regions of the silicon dioxide substrate.

A surface deposition sputtering coater (Japan Vacuum Instruments Ltd., MSP-1S) was employed to deposit a 15 nm Pt layer on the sidewalls of the V-grooves to preserve their morphology and composition. In situ TEM samples were prepared using focused ion beam (FIB) techniques. The sample and Cu grid were mounted on the stage and the height of the target area was adjusted to the eucentric height. Subsequently, a protective layer of approximately 1 µm Pt was deposited in the selected region under 30 kV voltage and 0.3 nA beam current. Following this, the region containing the V-groove was sectioned into a thin slice of 1.5–2 µm thickness using FIB, with the sample stage tilted by 7° to sever the bottom and sides of the slice. Subsequently, a 180° scanning rotation was used to secure the thin section onto a copper grid. The sample was then adjusted to the eucentric height and underwent a fine cleaning process to thin it down to approximately 100 nm. A larger beam current was initially used for speed, followed by a smaller beam current for precision. The sample was tilted at an angle of 52° with an accuracy of ± (0.5°–1.5°) to maintain a uniform thickness across the entire thin section. Finally, a low voltage of 2–5 kV was applied to minimize the thickness of the amorphous layer on both sides of the approximately 100 nm TEM (transmission electron microscope) sample section.

After the preparation of the TEM samples, an analysis was conducted on the nanogroove structure that was etched in six layers. Figure 5b shows its cross-sectional SEM image, while Figure 8a shows the cross-sectional TEM image. At a high resolution of 20 nm, Figure 8b displays the detailed cross-section of the nanogroove. The high-angle annular dark-field (HADDF) image in Figure 8c, with a resolution of 50 nm, reveals distinct compositional layering: The red dashed lines in the figure delineate distinct compositional layers, from left to right, which are Pt deposited during the FIB in situ TEM sample preparation, Pt deposited for sidewall protection after etching the nanogroove, SiO_2_, and Ga+, mainly originating from the FIB etching (due to Pt deposition after etching the nanogroove; the impact of Ga+ deposition during FIB TEM sample preparation on the initial sidewall composition is minimal). Figure 8d–h display the SEM EDS (energy dispersive spectroscopy) surface scan results, showing the elemental composition of the nanogroove sidewalls, which include Si, O, Pt, Ga, and C. Pt and Ga+ were deposited during the FIB preparation process, while C and O are common elements in the scanning electron microscopy chamber. The SiO_2_ sample contains Si and O. Pt was deposited in two separate steps, while Ga resulted from Ga+ deposition during both the nanogroove etching and the TEM sample preparation by FIB. The majority of Ga+ is expected to originate from the nanogroove etching process itself, as Pt deposition after etching would have minimized the impact of Ga+ deposition during FIB TEM sample preparation on the initial sidewall composition.

Under 20 nm high-resolution electron microscopy observation (Figure 8b), no micro-steps from layer-by-layer etching were observed on the sidewall morphology. This is attributed to the use of a 5 pA gallium ion beam (with a diameter of approximately 10 nm). During the etching process, most of the sputtered atoms were evacuated by the vacuum system, while a small fraction were redeposited on the sample surface and bottom. As the etching depth increased, the vacuum pump’s influence on the sputtered atoms decreased, intensifying the redeposition effect and resulting in the absence of observable micro-steps from layer-by-layer etching. Additionally, each theoretical step’s width is 10 nm, close to the ion beam diameter, which may have been disrupted during the etching process. The redeposition of sputtered atoms and Ga+ ions could also have filled these steps, making them difficult to observe.

This article also utilizes EDS line scanning to analyze the deposition from V-groove etching, with the results shown in Figure 9. Figure 9a, at a high resolution of 20 nm, performs an EDS line scan across the sidewall of the nanogroove from left to right, which includes Pt deposited during the FIB in situ TEM sample preparation, Pt deposited for sidewall protection after etching the nanogroove, and SiO_2_. Figure 9b illustrates the variation in elemental percentages with the scanning position. The elemental composition of the nanogroove sidewall includes Si, O, Pt, Ga, and C, and is consistent with the conclusions obtained from Figure 8. Figure 9b indicates that the boundary between the Pt layer and the SiO_2_ layer is approximately 55 nm from the starting point of the line scan. Assuming the deposit exists within a range of 55 nm ± 5 nm, the content of Si in this region is approximately 20%, O is about 65%, Ga is about 5%, and C is about 10%. Therefore, the etched deposit in the V-groove is predominantly composed of O, followed by Si and C, with a small amount of Ga, and is consistent with the conclusions obtained from Figure 8.

## 4. Influence of Etching Process Parameters on Nanostructures

### 4.1. Beam Current

Using a gold-coated SiO₂ substrate, a 30 kV FIB liquid metal Ga ion source was used to etch rectangles that were 200 nm wide and 2 μm long under different beam currents of 5 pA, 50 pA, 150 pA, and 200 pA. Fixed parameters included a 50% pixel overlap rate, 1000 scan passes, and a 500 μs dwell time. Measured through SEM images (with a scale bar of 1 μm), the depths of the rectangles were 345.6 nm, 1407.4 nm, 2477.4 nm, and 3193.4 nm, respectively, and the widths were 271.6 nm, 411.5 nm, 592.6 nm, and 789.3 nm, respectively. Figure 10e presents the curves showing how the rectangles’ depth and width vary with beam current size. As the beam current increases, the depth of the rectangles increases linearly, while the rate of width increase gradually slows. This is because the beam current size directly correlates with the number of incident ions; the greater the number of incident ions, the more material atoms are milled during the sputtering process. An increase in beam current leads to more ions interacting with the target material, thereby producing more sputtered atoms. The sputtering yield of these atoms directly impacts etching efficiency and depth. However, as the beam current increases, sputtered atoms redeposit on the sidewalls of the etched area, a phenomenon known as the redeposition effect. A higher beam current results in an increased sputtering rate, which in turn increases the likelihood of material redeposition. The redeposition effect slows down the increase in width because the redeposited atoms reduce the actual amount of material removed.

The specific experimental parameters and data are listed in Table 4.

### 4.2. Dwell Time

Tests were conducted on different dwell times using a gold-coated SiO_2_ substrate and an FIB liquid metal Ga ion source. The ion voltage was set to 30 kV and the current to 5 pA for etching five rectangles measuring 0.2 µm by 2 µm. Dwell times were set at 200 µs, 300 µs, 400 µs, 500 µs, 800 µs, and 1000 µs, with 1000 scans each, and a 50% pixel overlap rate in both the X and Y directions. The experimental results show that as the dwell time increases, both the depth and width of the rectangles increase. Specifically, from a dwell time of 200 µs to 1000 µs, the depth increased from 83.4 nm to 366.5 nm, and the width increased from 233.3 nm to 291.5 nm. For detailed data, refer to Table 5.

Figure 11e illustrates the trend in changes in the depth and width of the rectangle with dwell time. As dwell time increases, the depth of the rectangle increases significantly but at a gradually slowing rate, while the width increases more moderately. This is because an increased dwell time results in a higher number of ions, enhancing the milling effect and leading to greater etching depth. However, the increase in the aspect ratio of the rectangle hinders the sputtering of substrate atoms outward, increasing the redeposition effect, thereby slowing the rate of depth increase.

### 4.3. Number of Scans

Tests were conducted on different scan numbers using a liquid gallium ion source, a gold-coated SiO_2_ substrate, a voltage of 30 kV, and a current of 5 pA, while maintaining a 50% pixel overlap rate and a dwell time of 100 μs. By incrementally increasing the etching time, five rectangles of 0.2 μm by 2 μm were etched. The results show that as the number of scans increased from 2500 to 5000, the depth of the rectangles increased from 91.6 nm to 199.9 nm, while the width varied between 266.6 nm and 291.5 nm. For detailed information, see Table 6.

Figure 12e illustrates the variation in depth and width of the rectangle relative to the number of scans. As the number of scans increases, the depth of the rectangle gradually increases while the width remains almost unchanged, resulting in a relatively shallow overall etch depth. This is because the total etching time increases with more incident ions, leading to an increase in etching depth. However, due to the short dwell time, each individual etching pass achieves a small depth, thereby minimally affecting the overall morphology. Additionally, multiple passes can erode the layers that have been redeposited, which is why the total depth of etching is not very deep.

### 4.4. Pixel Overlap Rate

Under the same incident ion dose, experiments were conducted by adjusting the pixel overlap rate in the X direction (10%, 50%, 100%, 200%) while maintaining a 50% overlap rate in the Y direction and constant process conditions (liquid Ga ion source, gold-coated SiO_2_ substrate, 30 kV voltage, 5 pA current, 500 μs dwell time, 500 scans, 0.2 μm by 2 μm rectangular structure). Specific parameters and data can be found in Table 7.

The experimental results show that when the pixel overlap rate is below 100%, the bottom surface of the rectangle remains flat (Figure 13a–c). However, when the overlap rate reaches 200%, two grooves appear on the rectangle (Figure 13d). This phenomenon occurs because the pixel overlap rate is the ratio of the distance between the centers of adjacent ion beam spots to the diameter of the ion beam spot, which affects the effectiveness of the ion beam spots’ coverage. The beam profile follows a Gaussian distribution; at low overlap rates, the beam can cover the preceding pixel area, whereas at high overlap rates, the pixel spacing increases, preventing complete coverage and resulting in the formation of two grooves.

## 5. Conclusions

This study presents a layered etching method that overcomes the limitations of traditional focused ion beam (FIB) etching of nanogrooves. Leveraging the characteristics of Ga ion beams, we successfully fabricated V-grooves with a depth-to-width ratio of up to 9.6:1 and rectangular grooves with nearly perpendicular sidewalls. These grooves feature sharp apices for the V-grooves and serrated-free sidewalls, demonstrating the practicality and potential of this process. Through simulation of ion sputtering and redeposition processes and analysis of the effects of different process parameters on groove morphology and dimensions, the main conclusions are as follows:The comparison between the simulation results and experimental observations indicates that the sidewalls of the V-grooves in the experiment are smoother.The shaping effect of V-grooves closely resembles the simulation results, but actual widths exhibit deviations, primarily due to FIB focusing stability and redeposition effects. Redeposition causes sputtered atoms to redeposit on sidewalls, influencing etching depth and sidewall verticality. Smooth, saw-tooth-free sidewalls demonstrate the precise fabrication capability of FIB layering etching technology, effectively compensating for stratification caused by secondary ion sputtering.A line scan analysis revealed that the deposits are primarily composed of oxygen, followed by silicon, carbon, and a small amount of gallium. In the experimental preparation of rectangular grooves, the sidewall angle decreases with increasing etching width, approaching a near-vertical orientation.As the current increases, the depth of the rectangular grooves increases, and the opening width also increases, but at a slower rate of increase; with longer dwell times, the aspect ratio increases, and the width of the rectangles increases, but the rate of increase is relatively moderate; as the number of scans increases, the depth increases, but the overall depth is relatively shallow; the pixel overlap rate affects the morphology of the bottom surface, which is smooth when less than 100%, and double grooves appear when it reaches 200%.

## Figures and Tables

**Figure 1 micromachines-15-01105-f001:**
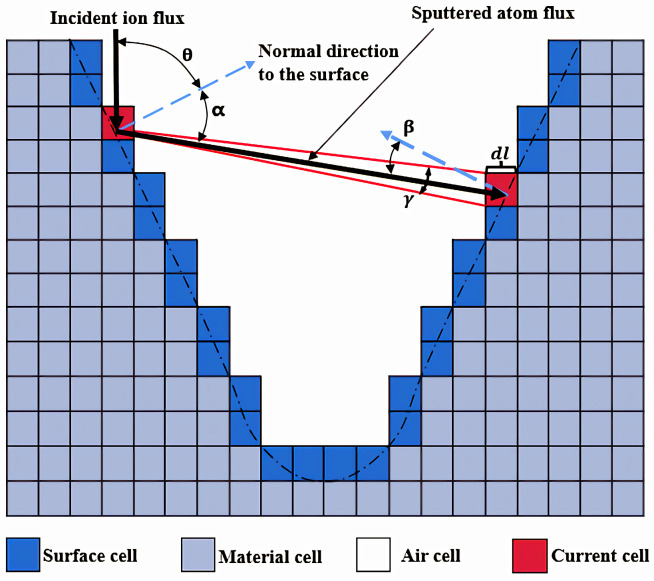
Schematic diagram of sputtered atom flux and redeposition flux calculation: θ represents the incidence angle of the ion beam, α is the emission angle of the sputtered atoms, β is the incidence angle of the redeposited atoms, dl is the edge length of the cell, and γ is the emission angle corresponding to the cell. “Surface cell”, “material cell”, and “air cell” indicate the current state of the cell, with the red grid representing the cells on the surface that are currently involved in sputtering and deposition.

**Figure 2 micromachines-15-01105-f002:**
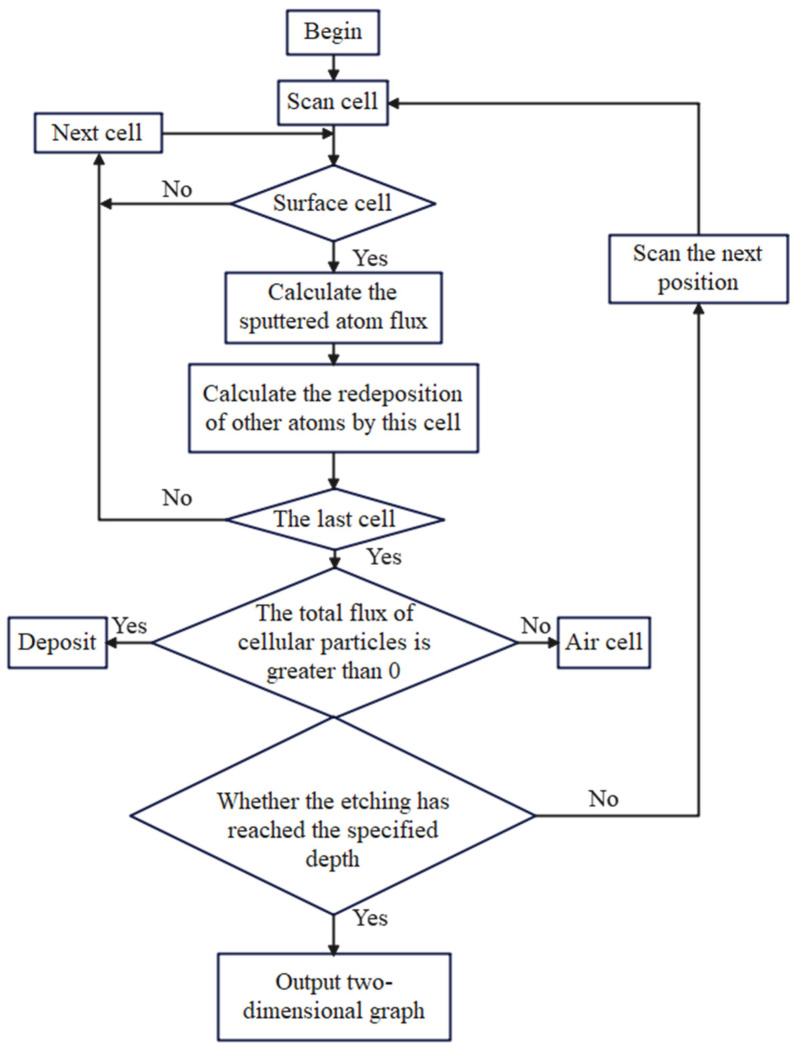
Flowchart of the cellular automaton simulation.

**Figure 3 micromachines-15-01105-f003:**
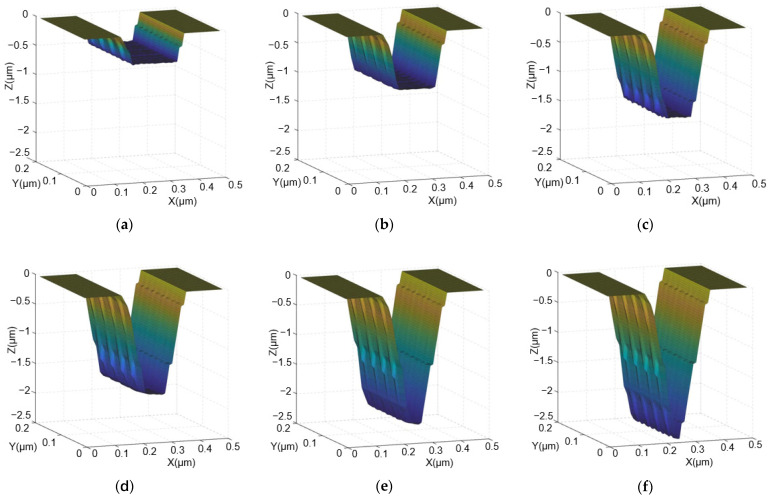
Flowchart of the cellular automaton simulation. The darker the color in the figure, the deeper the etching: (**a**) The three-dimensional surface diagram of the etched groove for two layers, with a width of 200 nm and a depth of 500 nm; (**b**) the three-dimensional surface diagram of the etched groove for four layers, with a width of 200 nm and a depth of 1000 nm; (**c**) the three-dimensional surface diagram of the etched groove for six layers, with a width of 200 nm and a depth of 1500 nm; (**d**) the three-dimensional surface diagram of the etched groove for seven layers, with a width of 200 nm and a depth of 1750 nm; (**e**) the three-dimensional surface diagram of the etched groove for nine layers, with a width of 200 nm and a depth of 2250 nm; (**f**) the three-dimensional surface diagram of the etched groove for ten layers, with a width of 200 nm and a depth of 2504 nm.

**Figure 4 micromachines-15-01105-f004:**
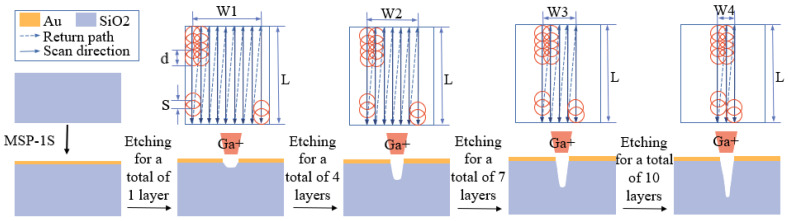
FIB layer-by-layer etching process diagram: The red circle represents the ion beam spot, d denotes the diameter of the ion beam spot, S signifies the spacing between spots, W indicates the total width of the scanning path, and L represents the length of the scanning path.

**Figure 5 micromachines-15-01105-f005:**
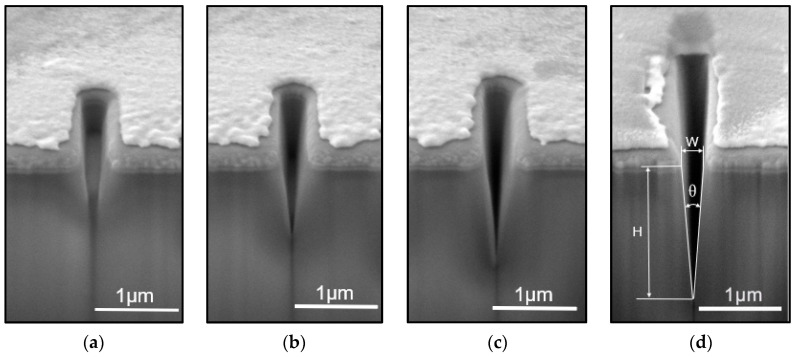
Cross-sectional SEM image of the layer-etched V-groove: (**a**) The structure after 4 layers of layer-by-layer etching, with a width of 242 nm and a depth of 388 nm; (**b**) the structure after 6 layers of layer-by-layer etching, with a width of 247 nm and a depth of 912 nm; (**c**) the structure after 8 layers of layer-by-layer etching, with a width of 245 nm and a depth of 1208 nm; (**d**) the structure after 10 layers of layer-by-layer etching, with a width of 175 nm and a depth of 1680 nm.

**Figure 6 micromachines-15-01105-f006:**
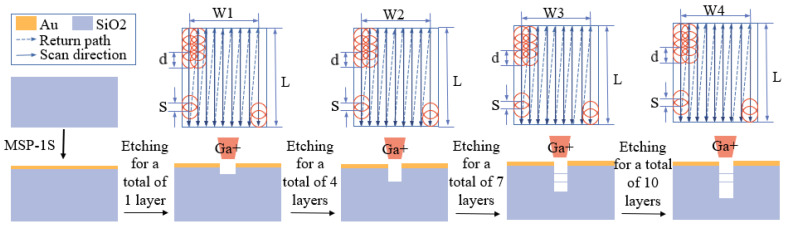
FIB layer-by-layer etching process diagram: The red circle represents the ion beam spot, where d denotes the diameter of the ion beam spot, S signifies the spacing between the spots, W indicates the total width of the scanning path, and L represents the length of the scanning path.

**Figure 7 micromachines-15-01105-f007:**
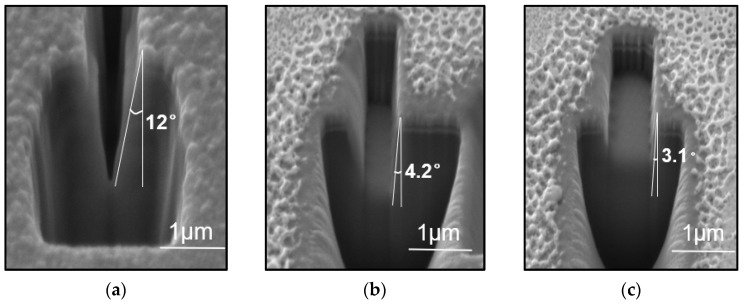
Cross-sectional SEM images of the layer-etched rectangular grooves: (**a**) For the structure with a set width of 200 nm, the actual width is 282 nm, the depth is 1072 nm, and the sidewall tilt angle is 12°; (**b**) for the structure with a set width of 400 nm, the actual width is 446 nm, the depth is 1000 nm, and the sidewall tilt angle is 4.2°; (**c**) for the structure with a set width of 600 nm, the actual width is 743 nm, the depth is 405 nm, and the sidewall tilt angle is 3.1°; (**d**) for the structure with a set width of 800 nm, the actual width is 892 nm, the depth is 396 nm, and the sidewall tilt angle is 2.8°. (**e**) The relationship between the etching width and the tilt angle of the groove sidewalls.

**Figure 8 micromachines-15-01105-f008:**
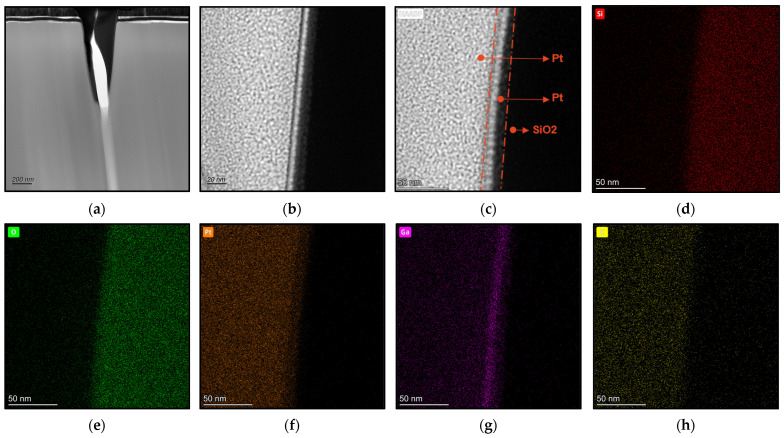
TEM analysis of V-groove tomography: (**a**) Cross-sectional TEM image of the nanogroove, with a resolution of 200 nm; (**b**) high-resolution cross-sectional TEM image of the nanogroove at 20 nm. (**c**) HAADF (high-angle annular dark-field) image from EDS line scan across the nanogroove section, with red dashed lines delineating distinct compositional layering: Pt deposited during FIB in situ TEM sample preparation, Pt deposited to protect the sidewalls after etching the nanogroove, and SiO_2_. (**d**–**h**) Elemental distribution maps obtained by EDS elemental mapping: (**d**) Silicon (Si), (**e**) oxygen (O), (**f**) platinum (Pt), (**g**) gallium (Ga), (**h**) carbon (C).

**Figure 9 micromachines-15-01105-f009:**
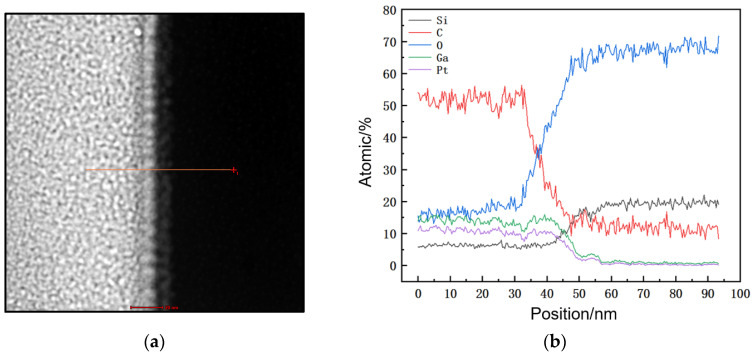
Line scan path and elemental distribution of the nanogroove: (**a**) The EDS line scan path along the left sidewall of the nanogroove; (**b**) the elemental distribution from the EDS line scan on the left sidewall of the nanogroove.

**Figure 10 micromachines-15-01105-f010:**
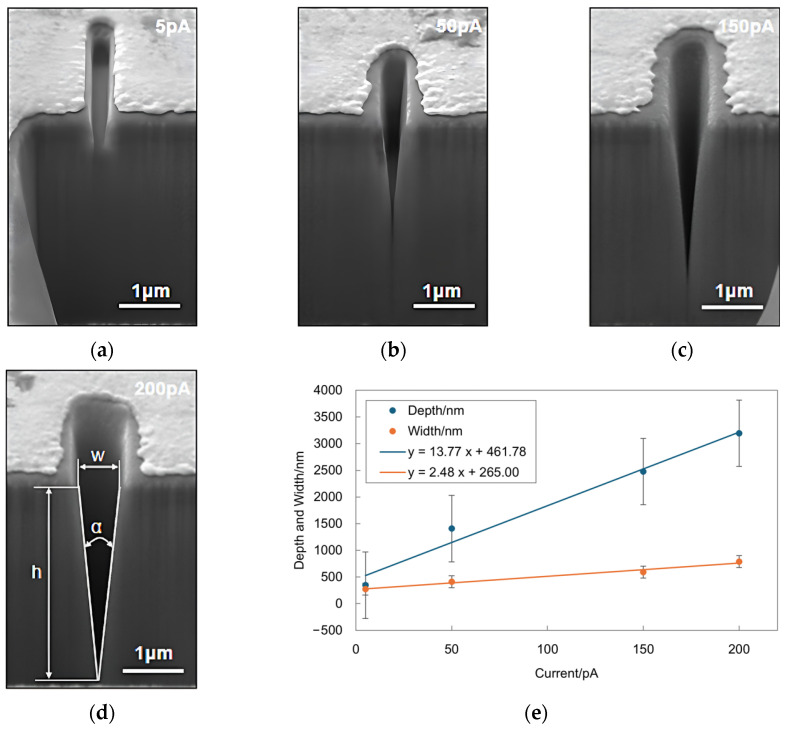
Contour morphology and depth–width dimension variations of nanostructures in experiments with different currents: (**a**–**d**) SEM cross-sectional images of 0.2 µm × 2 µm rectangles etched under the process conditions of an acceleration voltage of 30 kV, a pixel overlap rate of 50%, 1000 scan passes, a dwell time of 500 µs, and ion beam currents of 5 pA, 50 pA, 150 pA, and 200 pA, respectively. (**e**) Relationship curve between the depth and width of the rectangles and the ion beam current size.

**Figure 11 micromachines-15-01105-f011:**
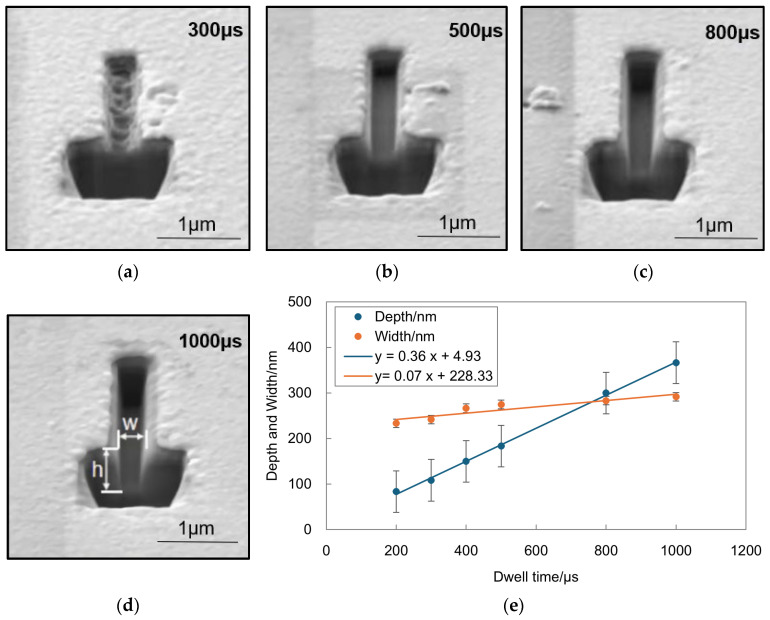
Contour morphology of nanostructures in experiments with different dwell times and changes in depth and width dimensions: (**a**–**d**) SEM cross-sectional images of the 0.2 μm by 2 μm rectangles etched under the conditions of an acceleration voltage of 30 kV, an ion beam current of 5 pA, a pixel overlap rate of 50%, and 1000 scan passes, with dwell times of (**a**) 300 μs, (**b**) 500 μs, (**c**) 800 μs, and (**d**) 1000 μs. (**e**) The relationship curve between the depth and width of the rectangle and the dwell time.

**Figure 12 micromachines-15-01105-f012:**
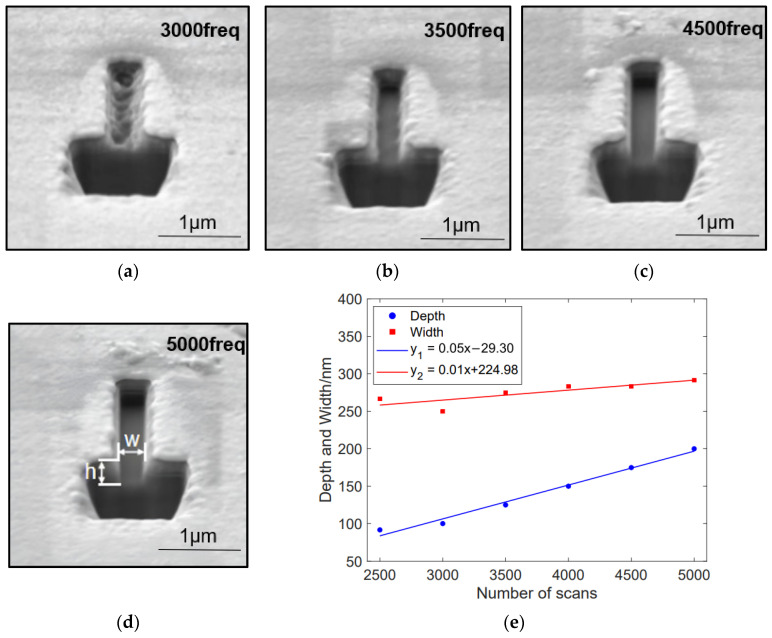
Changes in contour morphology and depth–width dimensions of nanostructures in experiments with different scan numbers: (**a**–**d**) SEM cross-sectional images of the 0.2 µm by 2 µm rectangles etched under the process conditions of an acceleration voltage of 30 kV, an ion beam current of 5 pA, a pixel overlap rate of 50%, and a dwell time of 100 µs, with scan numbers of (**a**) 3000, (**b**) 3500, (**c**) 4500, and (**d**) 5000, respectively. (**e**) The relationship curve between the depth and width of the rectangle and the number of scans.

**Figure 13 micromachines-15-01105-f013:**
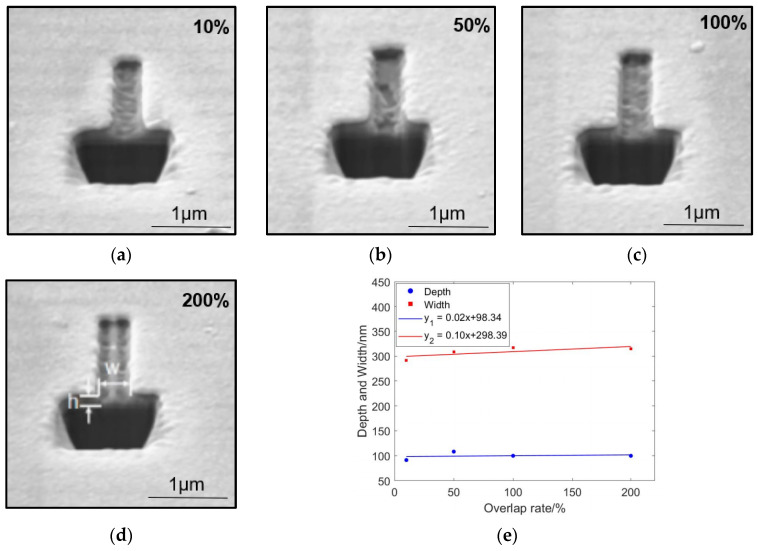
Changes in contour morphology and depth–width dimensions of nanostructures in experiments with different pixel overlap rates: Under an acceleration voltage of 30 kV, with an ion beam current of 5 pA, a pixel overlap rate of 50% in the Y direction, a dwell time of 500 μs, and 500 scan passes, SEM cross-sectional images were obtained for 0.2 μm by 2 μm rectangles etched under process conditions where the pixel overlap rates in the X direction were (**a**) 10%, (**b**) 50%, (**c**) 100%, and (**d**) 200%. (**e**) The relationship curve between the depth and width of the rectangle and the pixel overlap rate.

**Table 1 micromachines-15-01105-t001:** V-groove processing: Experimental parameters for each simulated etching step geometry.

Ion Voltage: 30 kV; Ion Current: 5 pA; Pixel Overlap Rate: 50%			
	Length (nm)	Width (nm)	Total Etching Time
Step 1 (Rectangle)	200	200	20 s
Step 2 (Rectangle)	200	180	20 s
Step 3 (Rectangle)	200	160	20 s
Step 4 (Rectangle)	200	140	20 s
Step 5 (Rectangle)	200	120	20 s
Step 6 (Rectangle)	200	100	20 s
Step 7 (Rectangle)	200	80	20 s
Step 8 (Rectangle)	200	60	20 s
Step 9 (Rectangle)	200	40	20 s
Step 10 (Rectangle)	200	20	20 s

**Table 2 micromachines-15-01105-t002:** Experimental parameters for each etching step geometry in V-pgroove processing.

Ion Voltage: 30 kV; Ion Current: 5 pA; Pixel Overlap Rate: 50%			
——	Length (nm)	Width (nm)	Total Etching Time
Step 1 (Rectangle)	200	200	20 s
Step 2 (Rectangle)	200	180	20 s
Step 3 (Rectangle)	200	160	20 s
Step 4 (Rectangle)	200	140	20 s
Step 5 (Rectangle)	200	120	20 s
Step 6 (Rectangle)	200	100	20 s
Step 7 (Rectangle)	200	80	20 s
Step 8 (Rectangle)	200	60	20 s
Step 9 (Rectangle)	200	40	20 s
Step 10 (Rectangle)	200	20	20 s

**Table 3 micromachines-15-01105-t003:** Experimental parameters for each etching step of rectangular groove processing.

—	——	Length (nm)	Width (nm)	Ion Current (pA)	Etching Time (s)	Etching Width (nm)	Etching Depth (nm)	Sidewall Tilt Angle (°)
1	Step 1 (Rectangle)	2000	200	5 pA	40	282	1072	12
	Step 2 (Rectangle)	2000	200	5 pA	60			
	Step 3 (Rectangle)	2000	200	5 pA	80			
2	Step 4 (Rectangle)	2000	400	5 pA	40	446	1000	4.2
	Step 5 (Rectangle)	2000	400	5 pA	60			
	Step 6 (Rectangle)	2000	400	5 pA	80			
3	Step 7 (Rectangle)	2000	600	5 pA	40	743	405	3.1
	Step 8 (Rectangle)	2000	600	5 pA	60			
	Step 9 (Rectangle)	2000	600	5 pA	80			
4	Step 10 (Rectangle)	2000	800	5 pA	40	892	396	2.8
	Step 11 (Rectangle)	2000	800	5 pA	60			
	Step 12 (Rectangle)	2000	800	5 pA	80			

**Table 4 micromachines-15-01105-t004:** Depth and width of the rectangle at different currents.

Ion Voltage: 30 kV; Pixel Overlap Rate: 50%; Number of Scans: 1000; Dwell Time: 500 µs		
Ion Current Magnitude/pA	Depth/nm	Width/nm
5	345.6	271.6
50	1407.4	411.5
150	2477.4	592.6
200	3193.4	789.3

**Table 5 micromachines-15-01105-t005:** Depth and width of the rectangle at different dwell times.

Ion Voltage: 30 kV; Ion Current: 5 pA; Pixel Overlap Rate: 50%; Number of Scans: 1000		
Dwell Time/µs	Depth/nm	Width/nm
200	83.4	233.3
300	108.3	241.6
400	149.9	266.6
500	183.3	274.9
800	299.9	283.2
1000	366.5	291.5

**Table 6 micromachines-15-01105-t006:** Depth and width of the rectangle at different number of scans.

Ion Voltage: 30 kV; Ion Current: 5 pA; Pixel Overlap Rate: 50%; Dwell Time: 100 µs		
Number of Scans/Time	Depth/nm	Width/nm
2500	91.6	266.6
3000	100.0	249.9
3500	124.9	274.8
4000	149.9	283.2
4500	174.8	283.2
5000	199.9	291.5

**Table 7 micromachines-15-01105-t007:** Depth and width of the rectangle at different pixel overlap rates.

Ion Voltage: 30 kV; Ion Current: 5 pA; Dwell Time: 500 µs; Number of Scans: 500 Times		
Pixel Overlap Rate/%	Depth/nm	Width/nm
10	91.4	291.6
50	108.3	308.3
100	99.9	316.6
200	99.9	314.5

## Data Availability

Data are contained within the article.

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
