# Peer review of "Controllable Fabrication of Gallium Ion Beam on Quartz Nanogrooves"

_micromachines, 2024, doi:10.3390/mi15091105_

Round 1

Reviewer 1 Report

Comments and Suggestions for Authors

The authors present a study on the optimization of Focused Ion Beam (FIB) techniques for the fabrication of nanogrooves in quartz. The simulation results were validated through experiments, and the samples were characterized using multiple measurement methods. While the experimental results deviated from the simulated expectations, they remained well within the error tolerance. The referee suggests that the editor consider accepting this manuscript for publication after addressing the following concerns:

1. Add non-breaking spaces between numbers and their units.

2. Ensure that figures/tables and their captions remain on the same page.

3. Provide references for the equations mentioned in the manuscript, except for those derived by the authors.

4. In the simulation model, the authors considered the Gaussian distribution of ion energy but assumed that the initial incident angle of the ions is 90° relative to the sample surface. This assumption is not rigorous. The incident angles of accelerated ions typically follow a distribution pattern influenced by the plasma generation power (doi: 10.3390/mi11090864). How does this assumption (ignoring the ion angular distribution) affect the final simulation result?

5. A thin layer of gold was deposited on the quartz sample before the FIB process. Clarify whether the gold in the etching area was removed before the FIB process or during it. If the gold was removed during the FIB process, even a thin layer could take time to remove, potentially affecting the final etching result, such as reducing the etching depth. Please provide clarification.

6. Regarding the gold conductive layer, it is known that this layer can alter the local electric field distribution and affect the incident angles of incoming ions (doi: 10.1016/j.apsusc.2022.152938). Please comment on how this might impact your etching results.

7. In Tables 1-3, the first column lists ‘Step 1 (rectangular) …’, but Tables 1 and 2 seem to discuss v-shaped grooves. Please double-check as this is confusing.

8. On page 15/22, the phrase ‘This thesis also utilizes EDS line scanning’ should be corrected, as this is not a thesis.

9. In Figures 10 and 11, add error bars to the measurement results.

10. Some of the SEM images are quite blurry. It is recommended to use clearer images.

Comments on the Quality of English Language

Minor errors are observed, and need careful check before publication.

Reviewer 2 Report

Comments and Suggestions for Authors

This work has demonstrated a layered etching method via FIB for constructing nano grooves. Simulations and experiments were designed and performed to verify the capacity and features of this method. The effect of the parameters was also explored in detail. The content was properly organized, and the experiment results were carefully discussed. Its suggested that this work can be accepted after proper revision.

1. Please adjust the focus of the introduction and enrich the recent progress on FIB etching.

2. The advantage of the layered etching method compared with the traditional one should be illustrated more clearly by data.

3. The pictures and tablets in this article are not uniform, such as Tablet 3, compared with others, and some figures with or without black frame lines.

4. Number, like 2.2 x 10^10 , 10^-7 , ..., should be rewritten in standard style.

Comments on the Quality of English Language

minor editing

Round 2

Reviewer 1 Report

Comments and Suggestions for Authors

The authors have addressed all concerns very effectively.